# Analysis of White Mulberry Leaves and Dietary Supplements, ATR-FTIR Combined with Chemometrics for the Rapid Determination of 1-Deoxynojirimycin

**DOI:** 10.3390/nu14245276

**Published:** 2022-12-10

**Authors:** Agata Walkowiak-Bródka, Natalia Piekuś-Słomka, Kacper Wnuk, Bogumiła Kupcewicz

**Affiliations:** 1Department of Inorganic and Analytical Chemistry, Faculty of Pharmacy, Nicolaus Copernicus University, 85-089 Torun, Poland; 2Department of Biostatistics and Biomedical Systems Theory, Faculty of Pharmacy, Nicolaus Copernicus University, 85-067 Torun, Poland

**Keywords:** mulberry, *Morus alba*, 1-deoxynojirimycin, DNJ, dietary supplements, PLS regression

## Abstract

Diabetes mellitus is a metabolic disease affecting more people every year. The treatment of diabetes and its complications involve substantial healthcare expenditures. Thus, there is a need to identify natural products that can be used as nutraceuticals to prevent and treat early-stage diabetes. White mulberry (*Morus alba* L.) is a plant that has been used in traditional Chinese medicine for thousands of years due to its many beneficial biological properties. White mulberry leaves are a source of 1-deoxynojirimycin (DNJ), which, due to its ability to inhibit α-glucosidase, can be used to regulate postprandial glucose concentration. In addition to consuming dried white mulberry leaves as herbal tea, many functional foods also contain this raw material. The development of the dietary supplements market brings many scientific and regulatory challenges to the safety, quality and effectiveness of such products containing concentrated amounts of nutraceuticals. In the present study, the quality of 19 products was assessed by determining the content of DNJ, selected (poly)phenols and antioxidant activity (DPPH^•^ assay). Nine of these products were herbal teas, and the other samples were dietary supplements. These results indicate the low quality of tested dietary supplements, the use of which (due to the low content of nutraceuticals) cannot bring the expected beneficial effects on health. Moreover, a method for determining the content of DNJ (the essential component for antidiabetic activity) based on ATR-FTIR spectroscopy combined with PLS regression has been proposed. This might be an alternative method to the commonly used chromatographic process requiring extraction and derivatization of the sample. It allows for a quick screening assessment of the quality of products containing white mulberry leaves.

## 1. Introduction

Diabetes mellitus (DM) is one of the most prevalent metabolism-related disorders associated with impaired insulin production (type 1) or developed insulin resistance (type 2). According to the International Diabetes Federation, in 2019, nearly 10% of people aged 20–79 (463 million) were living with diabetes worldwide, and this is expected to reach 700 million by 2045 [1]. The increased prevalence of diabetes mellitus means that cardiovascular disorders, blindness, stroke, kidney failure, foot ulcers, and depression will also increase. By 2045, the annual healthcare expenditure on treating diabetes and its complications are expected to be USD 845 billion [1]. Type 2 diabetes mellitus (T2DM) accounts for approximately 90% of all diabetic patients and is strongly associated with the current obesity epidemic [2]. Oral hypoglycemic drugs are the basis of pharmacotherapy for type 2 diabetes. Among them, α-glucosidase inhibitors (e.g., acarbose, voglibose, and miglitol) are recommended as the first-line therapy [3]. The α-glucosidase is an enzyme that catalyzes the hydrolysis of glycosidic bonds in dietary carbohydrates to absorb monosaccharides in the small intestine. Inhibiting the activity of this biomolecule reduces postprandial hyperglycemia, which plays a crucial role in the treatment and prevention of diabetes mellitus and its complications [4]. Although drugs from the group of α-glucosidase inhibitors have fewer side effects than other oral hypoglycemic drugs (sulfonamides, glinides, gliptins, biguanides, and thiazolidinediones), with prolonged use, they may cause some side effects such as gastrointestinal reactions and liver damage. [3]. Therefore, the search for plants/natural products that can be used as functional food, and identifying their active compounds that could be used as nutraceuticals or drugs has drawn considerable attention.

White mulberry (*Morus alba* L.) belongs to the mulberry family (*Moraceae*) and is native to Central Asia, but nowadays it is also cultivated in Europe. It has been used in traditional Chinese medicine for thousands of years. White mulberry leaves contain various beneficial components to health such as flavonoids, alkaloids, phenolics, amino acids, and polysaccharides. Numerous studies have shown that functional components included in white mulberry possess abundant biological activities, including antidiabetic, hypolipidemic, antiatherogenic, anticancer, cardiovascular, cardioprotective, antidopaminergic, antibacterial, antioxidant, and anti-inflammatory effects [5,6,7,8]. According to research presented in [9], the entire group of compounds (flavonoids and polysaccharides) found in white mulberry leaves is responsible for antidiabetic effects.

The main active mulberry component, 1-deoxynojirimycin (DNJ), is the polyhydroxylated piperidine alkaloid (tertrahydroxy piperidine derivative) (Figure 1), which can be defined as a glucose analogue with an amine group substitution for the oxygen atom in the pyranose ring [10]. DNJ competitively inhibits intestinal α-glucosidase and thus reduces glucose absorption, leading to lower blood glucose levels. In addition to its role in modulating glucose and insulin metabolism, DNJ also exhibits lipid-regulating and anti-obesity activity [11], inhibits adipogenesis [12], and is likely to have neuroprotective effects and may be an essential factor in preventing pathological brain changes in patients with Alzheimer’s disease [13].

The mechanism of DNJs multidirectional action still needs to be characterized. Detailed proteomic studies [14] show that long-term supplementation of mulberry leaf powder could enhance metabolic regulation by modulating the expression of signaling proteins in the insulin signaling pathway. Furthermore, cited work [14] also demonstrated an improvement in the functioning of mitochondria after supplementation of DNJ from mulberry leaves. Mitochondrial dysfunction leads to a decrease in ATP production and an increase in the production of reactive oxygen species, and consequently, to the development of insulin resistance. Improving mitochondria functioning is, therefore, a crucial element in preventing and treating type 2 diabetes. Other detailed studies on the mechanism of action of mulberry leaf extract and DNJ in mice suggested that it improved insulin resistance by modulating the insulin signaling pathway in the skeletal muscle of db/db mice after mulberry leaf or DNJ supplementation [15].

Interesting sources of DNJ are culture supernatant extracts (CSE) obtained from *Bacillus* sp. and *Streptomyces* sp. According to [16], appropriately selected culture conditions of *B. amyloliquefaciens* to allow obtaining SCE with the ability to lower postprandial glucose levels were comparable to white mulberry leaf extract. The literature indicates the potential use of DNJ from microorganisms for functional purposes [16].

From an analytical point of view, DNJ is a demanding molecule. Because of its high hydrophilicity and small molecular weight, the interaction with the stationary phase of conventional reverse-phase liquid chromatography (RP-HPLC) columns is so weak that DNJ is not retained in the column. The lack of chromophores in the DNJ structure (as in many other aminoglycosides) makes it impossible to use direct ultraviolet or fluorescence detection. To use RP-HPLC with a UV detector (the most widely used quantitative analysis technique) to quantify DNJ, it is necessary to derivatize the sample [17,18]. This procedure extends the analysis time, generates costs, and degrades the sample.

It is worth noting that some DNJ analogues, such as the miglitol mentioned above (*N*-EtOH-DNJ), miglustat (*N*-Bu-DNJ), and migalastat (DGJ, stereoisomer of DNJ), are registered drugs used for the treatment of type 2 diabetes, type I Gaucher disease or Niemann–Pick type C lysosomal storage diseases, and Fabry disease, respectively [19]. Importantly, research [20] suggests that miglitol can restore the counterregulatory response to hypoglycemia following antecedent hypoglycemia. In addition, data are available on other possible applications of these drugs, e.g., the treatment of cystic fibrosis [21], cancer [22], or COVID-19 [23]. In addition, the compound *N*-(5-adamantane-1-yl-methoxy-pentyl)-deoxynojirimycin (AMP-DNM) seems to be promising in the treatment of diabetes and obesity by promoting satiety, activating brown adipose tissue [24], and the effect on sterol regulatory element-binding proteins [25].

White mulberry leaves are most often consumed in teas or dietary supplements (tablets, capsules) containing extracts or powdered dried raw material. The global dietary supplements market size was USD 61.20 billion in 2020, estimated to be USD 128.64 billion in 2028 [26]. The increasing interest in a healthy lifestyle and well-being is expected to be a key driving factor for the dietary supplements market. The much lower requirements for the production process and quality control of dietary supplements compared to drugs, such as the fact that plant raw materials are not subject to standardization, with the high complexity of herbs and extracts that may cause severe problems with their quality and may be a potential threat to the consumer [27]. The diversification of the chemical composition of plant products may result from the origin of the plant, the time and place of harvesting, the drying process, extraction, the presence of impurities, or deliberate forgery. Therefore, the quality control of plant-origin samples presents several challenges for modern analytical chemistry. 

There are various spectroscopic techniques for a characteristic chemical profile of a plant or a fingerprint analysis. Fourier-transform infrared spectroscopy (FTIR) combined with multivariate data analysis is an effective tool for extracting specific chemical information. This approach has become a standard procedure for herbal species analysis. The application of attenuated total reflectance—Fourier transform infrared spectroscopy (ATR-FTIR) is strongly encouraged because it is rapid, non-destructive, and inexpensive compared to other analytical methods. The IR spectrum contains information on the biochemical composition of the sample, chemical bonds, and functional groups of the compounds. It allows for detecting differences between samples based on their chemical fingerprints [28,29]. 

This study aimed to quantify DNJ in white mulberry dry leaves (teas) and dietary supplements. To the best of our knowledge, there are no available data on determining the content of DNJ in products sold as dietary supplements. Moreover, the quality assessment of dietary supplements was based on determining the content of chlorogenic acid (CGA), neochlorogenic acid (nCGA), and rutin. Additionally, the DPPH^•^ radical scavenging activity was evaluated to measure antioxidant properties.

This work also presents the potential application of mid-infrared spectroscopy with ATR sampling combined with chemometric tools for relatively fast and non-destructive quantification of DNJ in white mulberry leaf products. 

## 2. Materials and Methods

### 2.1. Materials

All chemicals and solvents used in this study were purchased from Sigma–Aldrich (Saint Louis, MO, USA). The solvents (acetonitrile, water, methanol, and formic acid) used in this study were all analytical and HPLC grade. 2,2-diphenyl-1-picrylhydrazil (DPPH^•^) was used to prepare the solution with an absorbance value of about 1. 1-deoxynojirimycin (DNJ), chlorogenic acid, neochlorogenic acid, and rutin were used as analytical standards. Sodium borate buffer (pH 8.5), glycine, acetic acid, and 9-fluorenylmethyl chloroformate (FMOC-Cl) were used in the derivatization process of DNJ in samples. 

The study analyzed 19 herbal products containing white mulberry leaves or extracts. Ten of them (S1–S10) were dietary supplements. The other samples were herbal teas (T1–T9). The commercial products were purchased from local pharmacies and markets (Bydgoszcz, Poland) and online pharmacies. 

### 2.2. HPLC Analysis

High-performance liquid chromatography with a photodiode array detector (HPLC-DAD), (Shimadzu Corp., Kyoto, Japan) was used for the the quantitative analysis of 1-deoxynojirimycin, chlorogenic acid, neochlorogenic acid, and rutin in white mulberry leaves and dietary supplements samples.

For the preparation of samples for the determination of (poly)phenolic compounds (rutin, chlorogenic acid, neochlorogenic acid), about 500 mg of dried leaves or dietary supplement products (crushed in a mortar) were added to 15 mL of the methanol-water mixture (50:50), vortexed for 30 min, and then centrifuged (22,000 g) for 10 min (MPW-352RH centrifuge, MPW MED. INSTRUMENTS, Warsaw, Poland). The obtained extracts were filtered through a syringe filter with a pore diameter of 0.45 µm and subjected to further analysis.

A Kinetex^®^ column (150 mm × 4.6 mm, 5 µm) was used to separate (poly)phenolic compounds. The analysis was carried out using reverse-phase high-performance liquid chromatography (RP-HPLC). The mobile phase consisted of 0.2% formic acid in water (phase A) and acetonitrile (phase B). At a flow rate of 0.7 mL/min, the gradient was as follows: 0–2 min 10% B, 2–6 min 10–24% B, 6–11 min 10–24% B, 11–16 min 24–10% B and 16–25 min 10% B. UV detection wavelength of 325 and 350 nm and the injection volume of 20 μL was applied. Injection of each sample was performed in triplicate. The concentrations of rutin, chlorogenic acid, and neochlorogenic acid in samples were calculated from the calibration curves equation based on the peak area and recounted as the content in 1 g of dried white mulberry leaves.

The methods were validated for linearity, precision, accuracy, recovery, detection (LOD), and quantification (LOQ) limits according to ICH guideline ICH Q2 (R2) [30]. The detection limit was expressed as LOD = (3.3 · σ)/S, and the quantitation limit was defined as LOQ = (10 · σ)/S (where σ is the standard deviation of the response (n = 6), S is the slope of the calibration curve). The slope (S) was estimated from the calibration curve of the analyte, and σ based on the standard deviation of the blank. Recovery tests were completed by adding a known amount of the standard to the raw materials at two concentration levels. Then, the materials mixed with standards were prepared and analyzed under optimized conditions. Linearity was carried out on sets of standard solutions with different concentrations. The regression equations and regression coefficient (R^2^) values were calculated. Finally, precision was expressed as the relative standard deviation (%RSD).

The extraction and derivatization of samples for DNJ determination were carried out using the method described in [31]. About 500 mg of dried leaves or dietary supplement products (crushed in a mortar) wer added to 15 mL aqueous 0.05 M HCl, vortexed for 30 min, and centrifuged (22,000 g) for 10 min and subsequently filtered with a PTFE filter (0.45 µm pore size). The obtained extract was used for subsequent derivatization. Ten microliters of the extract was mixed with the same amount of 0.4 M sodium borate buffer (pH 8.5) in a microtube. Twenty microliters of 5 mM FMOC-Cl in acetonitrile was added with rapid mixing for 20 min at 20°C. To terminate the reaction by quenching the remaining FMOC-Cl, 10 μL of 0.1 M glycine was added. The mixture was diluted with 950 mL of 0.1% aqueous acetic acid (17.5 mM) to stabilize the DNJ-FMOC and filtered through a syringe filter (0.45 µm pore size).

DNJ was analyzed using a Kinetex^®^ column (150 mm × 4.6 mm, 5 µm) after derivatization with FMOC-Cl. Each peak was identified from a separate reaction with DNJ, glycine, or water. The detection wavelength was 265 nm. The samples were eluted with a mobile phase of 0.2% formic acid in water (phase A) and acetonitrile (phase B) with a ratio of 70:30 at a flow rate of 0.8 mL/min for 25 min. The injection of each sample was performed in triplicate. Method validation was performed analogously to the methods for (poly)phenolic compounds. The DNJ concentration in samples was calculated from the calibration curve equation based on the peak area and recounted as the content in 1 g of dried white mulberry leaves. The chromatograms of the derivatized standard DNJ and exemplary derivatized samples extracted from a dietary supplement and from mulberry leaves are shown in Appendix A.

### 2.3. Antioxidant Activity

This study used the DPPH^•^ radical (2,2-diphenyl-1-picrylhydrazyl) to assess antioxidant activity. The methanolic DPPH solution (0.05 M) was prepared and stored in darkness before the experiments. A volume of 150 μL methanolic-aqueous extracts (prepared as described in 2.2) was mixed with 1 mL DPPH^•^ methanolic solution. The mixture was shaken and incubated at 25 °C in the dark. Absorbance was measured after 30 min, at λ = 517 nm, against a blank. A Shimadzu model UV-VIS spectrophotometer (UV-1800) equipped with a quartz cell (10 mm optical path) was employed for the spectral measurements. Assays were performed in triplicate.

The EC_50_ (concentration of sample required to scavenge 50% of the DPPH^•^ free radicals) values were derived from the dose-response curve and recalculated to dry mulberry leaves. 

### 2.4. ATR-FTIR Spectroscopy

The ATR-FTIR experiments were carried out with a Ge-based ATR accessory (Pike Technologies, Madison, WI, USA) and a Shimadzu 8400 s spectrometer (Shimadzu Corp., Kyoto, Japan). Before the analysis, the solid samples (dry leaves, tablets, and capsules) were ground to a fine powder in a mortar. Next, a small portion of each sample was applied onto the surface of the ATR crystal and pressed by the clamp with constant pressure. The spectra (40 scans each) were measured in absorbance mode, within a 750–4000 cm^−1^ wavenumber range at a resolution of 4 cm^−1^. Each sample was measured in triplicate. 

### 2.5. Chemometric Analysis

The PLS (Partial Least Square) regression model with interval variable selection was constructed for a quantitative alternative to RP-HPLC and fast analysis of DNJ in samples. 

The data set was split using the Kennard–Stone algorithm into a calibration (15 products) and a test set (four products). Preprocessing was used, including baseline correction (weighted least squares baseline function) integrated with the standard normal variate method (SNV) and autoscaling. Variables were selected based on the interval PLS algorithm (iPLS), which is very useful for spectral data analysis [32]. The following three intervals were chosen: 1101–1196 cm^−1^, 1333–1427 cm^−1^, and 1603–1697 cm^−1^. Variable selection was processed in stepwise forward mode (size of interval = 30, variable step = 10, number of intervals = auto). 

The accuracy of the constructed model was assessed based on validation parameters: determination coefficient of calibration (R^2^_CAL_) and root mean square error of calibration (RMSE_CAL_). Cross-validation was employed as an internal validation using the leave-one-out method to assess how well the model described the relationships within the calibration data. Statistical outputs were calculated as follows: coefficient of determination of cross-validation (R^2^_CV_) and root mean square error of cross-validation (RMSE_CV_).

To evaluate the predictive ability of the model, an external test set was used, consisting of samples from four products (two dietary supplements and two teas). Using the test set, validation parameters were calculated, such as the determination coefficient of prediction (R^2^_PRED_) and root mean square error of prediction (RMSE_PRED_). Additionally, based on the William’s plot (distribution of studentized residuals against the leverage), the models applicability domain (AD) was determined. Furthermore, the Y-randomization test examined the verification of model robustness with a 100-response permutation. 

All the calculations were performed using PLS-Toolbox 7.5 (Eigenvector Research, Inc., Manson, WA, USA) and MATLAB software version R2020b (The MathWorks, Inc., Natick, MA, USA).

## 3. Results and Discussion

The use of functional food, including dietary supplements, can bring the expected beneficial effects only if their good quality is ensured. The lack of uniform unambiguous requirements for this type of nutrition is associated with a high risk of low-quality products in the market. For the above reasons, to assess the quality of products containing white mulberry leaves (in the form of herbal teas and dietary supplements), the content of DNJ (the essential biomolecule for the antidiabetic activity), (poly)phenolic compounds (chlorogenic acid, neochlorogenic acid, and rutin), and general antioxidant activity were determined.

### 3.1. Determination of DNJ, CGA, nCGA, and Rutin

In our study, we have used the HPLC-DAD technique to obtain a quantitative evaluation of DNJ, CGA, nCGA, and rutin content in white mulberry leaves and dietary supplements. The methods developed for this purpose are of high quality, as evidenced by the values of validation parameters presented in Table 1.

The optimization of an analytical method with adequate validation parameters is particularly important in quantifying DNJ due to the need to derivatize the sample. The calculated metrics demonstrate the high efficiency of derivatization and extraction processes. 

Figure 2 shows the results of DNJ determination, recalculated to the content (µg) per 1 g of dry leaves in white mulberry dietary supplements (S1–S10) and herbal teas (T1–T9). Since the DNJ values differ significantly, a gap on the scale was used for better visualization and comparison of the results (the gap was marked with the red dotted line). 

DNJ concentrations in dietary supplements (S1–S10) and leaves (T1–T9) (expressed as the content of DNJ in 1 g of dried leaves) were found to vary from 0.006 to 992.314 µg/g. The DNJ content in teas was similar and fluctuated in the range of 503.405 to 992.314 µg/g. These results are comparable with the literature [31,33,34,35]. An extremely low amount of DNJ was observed in the samples of dietary supplements (from 0.006 to 1.064 µg/g). Two products (S2 and S5) with the highest content of DNJ (but still very low) in the group of dietary supplements comprised of powdered white mulberry leaves. Other dietary supplements are produced using extracts of varying concentrations (drug extract ratio, DER from 4:1 to 20:1). There are many possible explanations for the low content of DNJ in dietary supplements. One of them is the low efficiency of the extraction process. The extraction yield is influenced by many factors, such as preparation of the raw material, type of solvent used, and duration and temperature of the process [36]. Inadequately selected conditions for extraction may lead to limited isolation of active substances or, on the other hand, their degradation. Lack of optimization of the extraction method may result in poor quality of the produced dietary supplements. Likewise, the formulation of dietary supplement tablets or capsules can negatively affect the active ingredient content. A disturbing possible explanation for the obtained results is the use of a lower amount of raw material than what was declared.

The literature has provided information on the recommended amounts of DNJ to lower postprandial glucose levels [37,38,39]. Suggested doses are in the range of 6–18 mg three times a day. Using 30 mg as a recommended dietary intake (RDI) and considering the number of tablets/capsules proposed by manufacturers, we calculated the percentage of the daily amount satisfied by taking dietary supplements (Figure 3). 

As seen in Figure 3, the use of any analyzed supplement, as suggested by the manufacturer, does not provide even 3% of the recommended dietary intake of DNJ.

The contents of (poly)phenolic compounds in the studied products are shown in Figure 4. The results are consistent with those obtained for DNJ determinations. In products with the status of dietary supplements, the content of (poly)phenolic compounds, responsible mainly for the antioxidant properties of white mulberry leaves, is negligible.

The available literature [40,41,42,43,44] indicate high variability in dried white mulberry leaf CGA, nCGA, and rutin content. As the authors emphasize, this is due to the differences between raw materials from various crops and the considerable influence of the type of extractant used. Nevertheless, the analysis performed in this study for tea products is consistent with previously published data, even for T5 and T7 products with the lowest (poly)phenol content.

Besides the very low content of CGA, nCGA, and rutin in studied dietary supplements, attention is also drawn to their changed proportion compared to tea samples. Half of the tested dietary supplements practically do not contain CGA, while in the case of teas, this compound was present in the largest amount among those determined. A relatively low content of rutin is also noticeable and is very similar in all tested dietary supplements containing the extract (S1, S3, S4, S6–S10). The above observations suggest a particular sensitivity of rutin and CGA to inappropriate extraction conditions (e.g., using too much ethanol, because it is easier to evaporate than water).

The chromatographic analyses indicate the low quality of the tested dietary supplements. Significant differences between the content of the active substances (DNJ, CGA, nCGA, and rutin) between teas and dietary supplements suggest using a smaller amount of the raw material than declared. On the other hand, higher contents of the active compounds were in only two dietary supplements (S2 and S5), which composed of dried white mulberry leaves. These results indicate an inaccurate optimization method for obtaining plant extracts (which are components of other dietary supplements).

### 3.2. DPPH^•^ Assay

Oxidative stress plays a critical role in diabetes and many other serious conditions, including aging, cancer, chronic inflammation, neurodegenerative diseases, atherosclerosis, etc. Therefore, plant materials showing antioxidant activity arouse the interest of researchers. Naturally occurring nutrients with potent antioxidant properties are (poly)phenols [45] found in dried white mulberry leaves. A commonly used method in assessing antioxidant properties (as free radical scavenging) is the DPPH^•^ assay. The results of the evaluation of free radical scavenging capacity for white mulberry dietary supplements and teas are presented in Figure 5. The lower the EC_50_ value, the greater the antioxidant capacity of the product. Since the EC_50_ values differ significantly, gaps on the scale were used for better visualization and comparison of the results (the gaps were marked with the red dotted lines). 

As can be seen in Figure 4, the free radical scavenging activity of teas (T1–T9) significantly exceeds other tested products (S1–S10). Among the dietary supplements, samples S2 and S5 exhibited greater antioxidant capacity than the others (middle part of Figure 5). These results are consistent with the content of nCGA, CGA, and rutin (Figure 4). The highest amounts of rutin and phenolic acids among dietary supplements were found in S2 and S5 products. As noted, these two dietary supplements contain powdered dried leaves of white mulberry, not an extract as in the others.

### 3.3. ATR-FTIR Spectra

The ATR-FTIR spectra of the investigated food products (Figure 6) displayed typical vibrational patterns of plant constituents, such as sugars, proteins, and lipids. White mulberry leaves contain about 15–30% proteins, 2–8% lipids, 10–40% carbohydrates, and 10–37% neutral dietary fiber [46]. The ATR-FTIR spectra of the mulberry leaves (blue) and dietary supplements containing extract from leaves (red) showed the broad and intense band at 2790–3570 cm^−1^ assigned to the stretch vibration of O-H and C-H, and it was not helpful in this work because all spectra were very similar in this range. In this region, 900–1700 cm^−1^ signals of the flavonoid molecules and DNJ can be found [47], which is confirmed by the spectra of rutin, chlorogenic acid, and DNJ analytical standards. The vibrations observed in 1650–1580 cm^−1^ can be attributed to N-H of amines. The peaks at 1572 and 1541 cm^−1^ are assigned to the aromatic ring (C-C) skeletal vibrations. The band at 1410 cm^−1^ was assigned to C-H bending vibration. Absorptions between 1300 and 1000 cm^−1^ showed stretching vibrations of the pyranose ring [31]. The absorption at 935 and 1205 cm^−1^ were assigned to C-C, C-O stretching, and C-O-H, C-O-C deformation modes of oligo- and polysaccharides modes [48]. The distinctive band at 1047 cm^−1^ can be assigned to the vibrational frequency of -CH_2_OH groups of carbohydrates [49]. Bands in the range of 1300–1492 cm^−1^ are associated with O-C-H, C-C-H, and C-O-H bending modes.

### 3.4. iPLS Model

The low content of DNJ, the essential component of white mulberry leaves, in the tested dietary supplements proves how important their quality control is. The relatively complicated and time-consuming procedure of quantitative analysis of DNJ (extraction and derivatization) prompted us to search for an alternative method based on spectroscopic techniques combined with chemometrics.

We decided to calculate the interval partial least squares (iPLS) regression model to predict DNJ content in samples containing white mulberry leaves based on the ATR-FTIR spectra. Appendix A shows the average spectrum of tested samples (dietary supplements and teas) and the DNJ spectrum after preprocessing using the SNV with highlighted intervals used to build the iPLS model. The model was built by employing three intervals (1101–1196 cm^−1^, 1333–1427 cm^−1^, and 1603–1697 cm^−1^) from the fingerprint region of FTIR spectra 850–1800 cm^−1^. Figure 7 presents the relationships between measured and predicted values of DNJ concentration using the model. The model, constructed with six latent variables (LV), had good validation metrics: RMSE_CAL_ = 0.025; RMSE_CV_ = 0.095; RMSE_PRED_ = 0.016; R^2^cal = 0.995, R^2^cv = 0.925 without overfitting. The permutation test (Y randomization) confirms the models good quality (all three tests: Wilcoxon, sign test, and Rand t-test, were passed, *p* < 0.05). Therefore, ATR-FTIR spectra combined with PLS regression can be an efficient tool for the prediction of DNJ in white mulberry leaf products (teas and dietary supplements).

## 4. Conclusions

The results of the quantitative chromatographic analysis indicated that dietary supplements with white mulberry leaf extract contained a negligible quantity of active substances, such as (poly)phenols and DNJ (α-glucosidase inhibitor), compared to the declared amount of raw material (dried leaves). These observations confirm the need to increase plant-based functional food quality control.

Herein, we propose an alternative to the chromatographic method for quantifying DNJ in food samples based on ATR-FTIR spectroscopy combined with PLS regression. This methodology does not require sample extraction and derivatization. Furthermore, it is relatively simple, non-destructive, inexpensive, and in line with green chemistry. Following the global need for appropriate regulation of dietary supplements, the method developed in the present study can be considered promising for routine quality control of dietary supplements containing white mulberry leaves.

## Figures and Tables

**Figure 1 nutrients-14-05276-f001:**
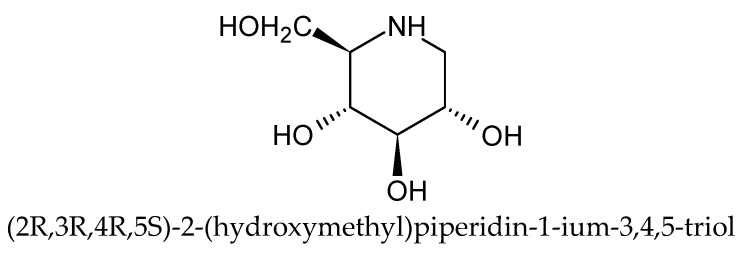
Structure of 1-deoxynojirimycin (DNJ).

**Figure 2 nutrients-14-05276-f002:**
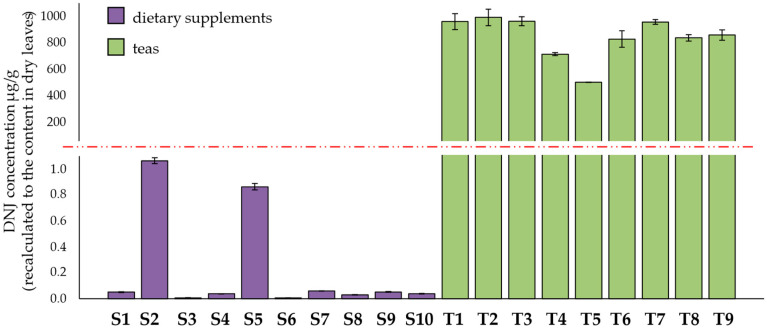
DNJ content in samples expressed in µg/g (recalculated to the content in dry leaves). The red dotted line indicates the gap on the scale (for DNJ concentration between 1.2–100 µg/g) .

**Figure 3 nutrients-14-05276-f003:**
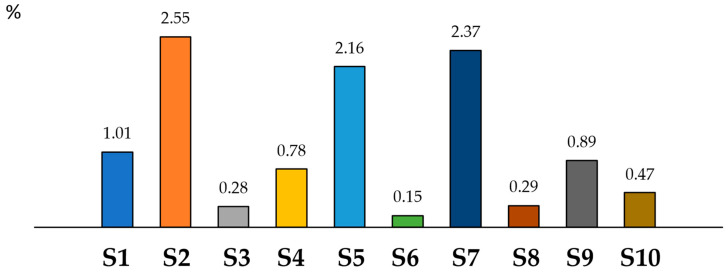
The percentage of recommended dietary intake of DNJ, which is covered by the dose of dietary supplements (S1–S10) suggested by manufacturers.

**Figure 4 nutrients-14-05276-f004:**
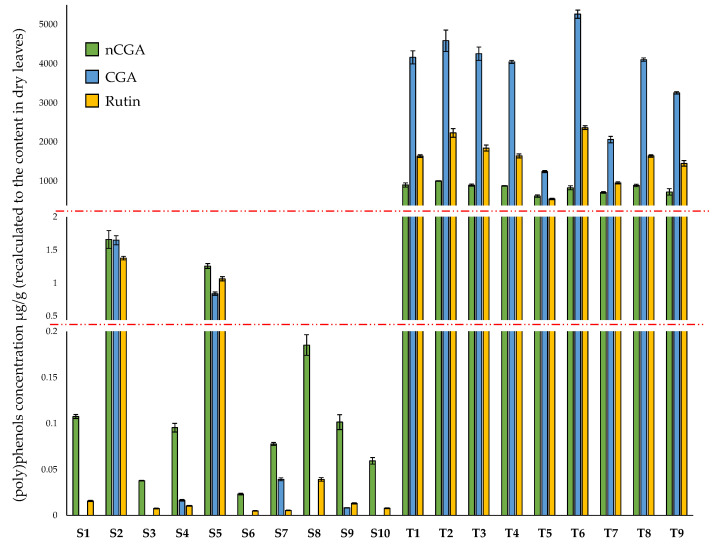
Chlorogenic acid (CGA), neochlorogenic acid (nCGA), and rutin content in samples expressed in µg/g (recalculated to the content in dry leaves). The red dotted lines indicate the gaps on the scale, for (poly)phenols concentration between 0.2–0.5 µg/g and between 2–400 µg/g).

**Figure 5 nutrients-14-05276-f005:**
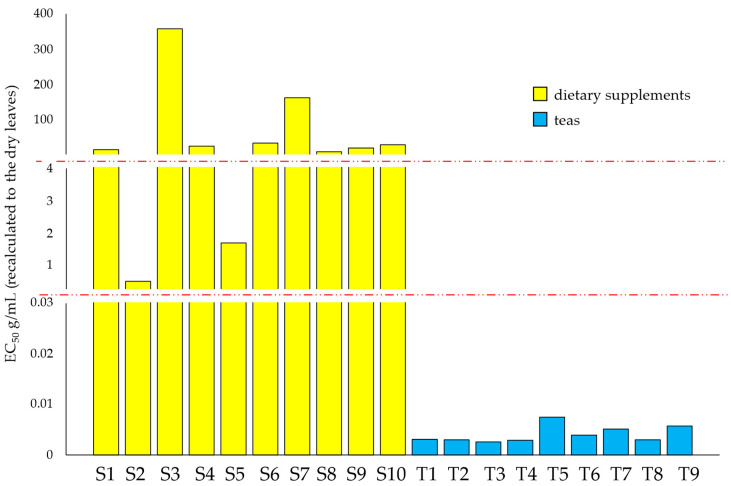
Results of DPPH^•^ assays (as EC_50_) of white mulberry teas and dietary supplements (recalculated to the dry leaves). The red dotted lines indicate the gaps on the scale, for EC50 values between 0.03–0.1 g/mL and between 4–10 g/mL).

**Figure 6 nutrients-14-05276-f006:**
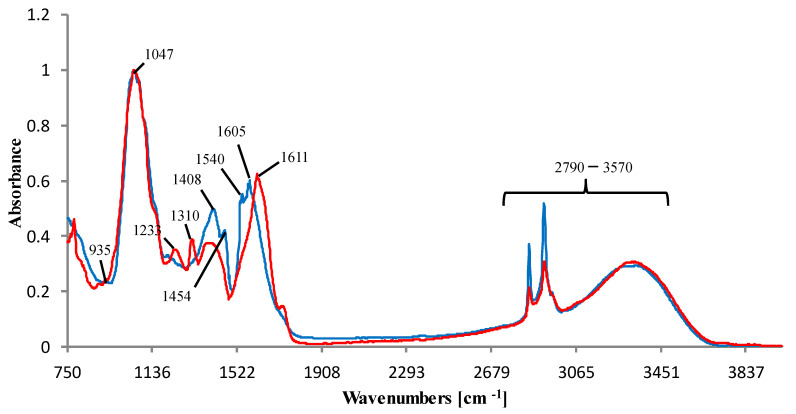
Normalized ATR-FTIR spectra of exemplary samples: white mulberry leaves (blue) and dietary supplement (red).

**Figure 7 nutrients-14-05276-f007:**
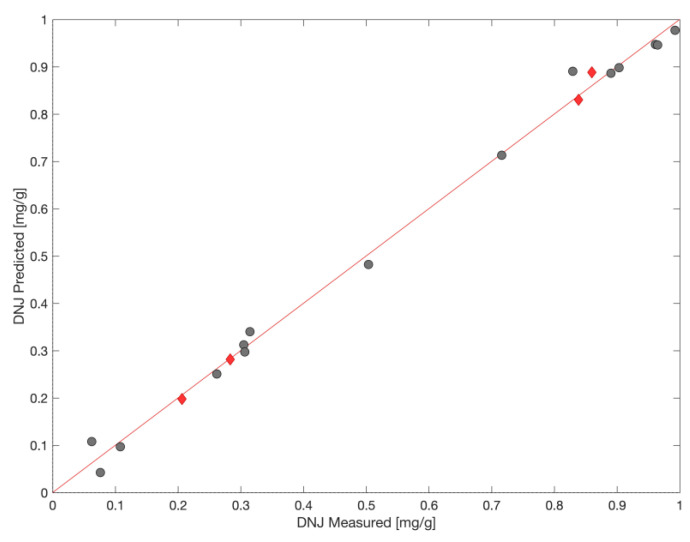
Results of iPLS model. Relationships between measured and predicted values of DNJ concentration using the model (●—calibration set, ♦—test set). LV number = 6; cross-validation method: leave-one-out; RMSE_CAL_ = 0.025; RMSE_CV_ = 0.095; RMSE_PRED_ = 0.016; R^2^_CAL_ = 0.995.

**Table 1 nutrients-14-05276-t001:** Validation metrics of RP-HPLC methods.

Parameter	DNJ	CGA	nCGA	Rutin
Range (µg/mL)	3.14–157.14	10.00–140.00	5.00–80.00	5.00–50.00
Regression equation	y = 88.68x − 0.06	y = 44.56x − 0.17	y = 52.56x − 0.10	y = 24.48x − 0.02
Regression coefficient (R^2^)	0.999 ± 0.001	0.995 ± 0.002	0.999 ± 0.001	0.999 ± 0.001
Recovery (%)	90.4 ± 1.7	96.2 ± 1.2	94.1 ± 0.6	93.8 ± 1.1
Precision (%RSD)	7.5	5.6	6.7	6.2
LOD (ng/mL)	9.1	3.52	2.82	4.07
LOQ (ng/mL)	27.7	10.67	8.55	12.33

Abbreviations: CGA, chlorogenic acid; DNJ, 1-deoxynojirimycin; LOD, limit of detection; LOQ, limit of quantification; nCGA, neochlorogenic acid; RSD, relative standard deviation.

## Data Availability

Not applicable.

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
