# Peer review of "Analysis of White Mulberry Leaves and Dietary Supplements, ATR-FTIR Combined with Chemometrics for the Rapid Determination of 1-Deoxynojirimycin"

_nutrients, 2022, doi:10.3390/nu14245276_

Round 1

Reviewer 1 Report

It is a good study and the topic is much needed. please 1) check and modify some phrases to make sure their accuracy, such as 'civilization disease'; 2) some statements need to be clear, such as "All chemicals and solvents were..." should be "All chemicals and solvents used in this study were..."; 3) because these products were not only specified by one particular compounds, other compounds should be examined as well, such as other polyphenols (flavonoids), in particular, stilbene compounds that are rich in mulberry leaves.

Author Response

Responses to the reviewer's comments are included in the attached file.

Reviewer 2 Report

This manuscript describes a study on the amount of dietary supplements and teas used for treatment of diabetes.

In particular the authors have studied the content of polyphenols and of 1-desoxy-norjirimycin, a small piperidine derivative inhibitor of alpha-glucosidase thus effective in preventing too rapid glucose formation from food carbohydrates.

The paper is well written and discuss the extraction methodology and the method for quantifying the principal component of each extract.

Then the authors show result of measurements in 10 food supplements and 10 teas from the market.

They show convincingly that most of the supplements do not bring enough active compounds for an effect. And that the teas are more similar in active ingredient content and content much higher amounts as the supplements.

They also develop a simple ATR-FTIR method to analyse quickly the food supplements and teas. They show that this method is able to report a correct value of 1-norjirimycin  and could be used to sort more or less efficient supplements or teas.

To help the reader, I think it would be nice to show the structure of 1-norjirimycin in a figure.

It would be nice to show figures of the HPLC chromatogram of extracts and the standard derivatized 1-norjirimycin (perhaps in a supplementary figure)

Page 2 line 62 : why not say it is the alkaloid a tertra hydroxypiperidine derivatrive ((2R,3R,4R,5S)-2-(Hydroxymethyl)piperidine-3,4,5-triol). And put an image of the molecule somewhere.

Page 8 line 295 : (e.g., using too much, 295 easier to evaporate, ethanol than water).

reformulate (e.g., using too much ethanol, because it is easier to evaporate than water). )

Page 9 fig 4 : It would be nice to show a IR spectrum of DNJ on the same diagram (for instance above it.) In order to show the chemical chromophores of the molecule? It is amazing that a statistical method is able to get quantitative data from spectra looking like potatoes.

All together it is a well written and clear manuscript and may be interesting to some health food addicts.

There are a few improvements possible as suggested.

Author Response

(The authors gave the same response as above.)
